# Cohort profile: Oxford Pain, Activity and Lifestyle (OPAL) Study, a prospective cohort study of older adults in England

Maria T Sanchez Santos ![ORCID],[1] Esther Williamson ![ORCID],[1] Julie Bruce ![ORCID],[2] Lesley Ward,[1,3] Christian D Mallen,[4] Angela Garrett,[1] Alana Morris,[1] Sarah E Lamb,[1,5] on behalf of the OPAL study team

For numbered affiliations see end of article.

**Correspondence to**
Professor Sarah E Lamb;
s.e.lamb@exeter.ac.uk

## ABSTRACT

**Purpose** The 'Oxford Pain, Activity and Lifestyle' (OPAL) Cohort is a longitudinal, prospective cohort study of adults, aged 65 years and older, living in the community which is investigating the determinants of health in later life. Our focus was on musculoskeletal pain and mobility, but the cohort is designed with flexibility to include new elements over time. This paper describes the study design, data collection and baseline characteristics of participants. We also compared the OPAL baseline characteristics with nationally representative data sources.

**Participants** We randomly selected eligible participants from two stratified age bands (65–74 and 75 and over years). In total, 5409 individuals (42.1% of eligible participants) from 35 general practices in England agreed to participate between 2016 and 2018. The majority of participants (n=5367) also consented for research team to access their UK National Health Service (NHS) Digital and primary healthcare records.

**Findings to date** Mean participant age was 74.9 years (range 65–100); 51.5% (n=2784/5409) were women. 94.9% of participants were white, and 28.8% lived alone. Over 83.0% reported pain in at least one body area in the previous 6 weeks. Musculoskeletal symptoms were more prevalent in women (86.4%). One-third of participants reported having one or more falls in the last year. Most participants were confident in their ability to walk outside. The characteristics of OPAL Cohort participants were broadly similar to the general population of the same age.

**Future plans** Postal follow-up of the cohort is being undertaken at annual intervals, with data collection ongoing. Linkage to NHS hospital admission data is planned. This English prospective cohort offers a large and rich resource for research on the longitudinal associations between demographic, clinical, and social factors and health trajectories and outcomes in community-dwelling older people.

## INTRODUCTION

The population of the UK is undergoing a fundamental change in its age structure, due to lower birth rates and extended life expectancy. One in four people in the UK are projected to be aged 65 or over by 2050, with

## Strengths and limitations of this study

► Oxford Pain, Activity and Lifestyle (OPAL) is a new, high-quality cohort of older community-dwelling people aiming to explore causes and consequences of pain, frailty, mobility decline, disability and poor health-related quality of life.

► A total of 5409 older adults from 35 general practices in nine distinct areas in England participated at baseline, 2016–2018.

► OPAL participants are similar to those in general population of the same age.

► The cohort study relies on self-reported and routine National Health Service (NHS) data, there is not face-to-face data collection.

► Our findings may under represent older people living in the community with severe cognitive impairment.

15% aged over 75 years and 5% aged 85 years or older.[1] This change reflects gains in health and social development, and it is important that as many years of life are spent in good health as possible.

Active independence is one of the key concerns of older people, and mobility is critically important for independence.[2 3] Older people value their mobility highly and consider mobility loss as a key disadvantage of ageing.[4] Poor or limited mobility is linked to functional decline, mortality and increased healthcare utilisation.[5] Conceptually, factors associated with mobility decline precede disability within models of disablement. Therefore, identification of factors associated with mobility decline is important for prevention of, and rehabilitation from, mobility decline.[6]

Musculoskeletal pain is one of the leading causes of disability and disease burden worldwide among community-dwelling older adults.[7 8] A recent review estimated that the

prevalence of chronic pain among older adults in the UK ranged from 42% in 65–74 years old to 62% in the over 75 age group.[9] These prevalence estimates are similar to other developed countries.[10]

Musculoskeletal pain has a large impact on many other aspects of older people's health such as loss of mobility, frailty, cognitive impairment, falls and poor sleep quality.[11–15] However, the role of musculoskeletal pain on adverse health outcomes in older adults is poorly understood. The majority of studies are cross-sectional in design, thus are limited; and only few longitudinal studies have examined potential mediators between pain and disability.[16] A better understanding of the causal path between musculoskeletal pain and disability in representative community-based older adults is needed to inform decisions about treatment and rehabilitation.

There are a number of high-quality cohort studies examining age-related health conditions among community-dwelling older adults. These include the English Longitudinal Study of Ageing (ELSA), the Maintenance of Balance, Independent Living, Intellect, and Zest in the Elderly of Boston Study (MOBILIZE), the Longitudinal Study of Ageing, the Baltimore Longitudinal Study of Aging and the Italian Invecchiare in Chianti Study (InChianti), among many others. However, to our knowledge, only one cohort focuses on the impact and contribution of musculoskeletal pain on disability in older people, the ongoing MOBILIZE Boston Study.[17] This American Cohort is limited by a relatively small sample size (765 participants at inception).

In order to address these knowledge gaps, we assembled the Oxford Pain, Activity and Lifestyle (OPAL) Cohort, a prospective study of community-dwelling older adults from across England. The immediate objectives were:

► To investigate the causes and consequences of mobility decline and disability in later life, and the role and contribution of musculoskeletal pain and other factors.
► To develop a prognostic tool to assess mobility decline in a population-based cohort of older adults in UK.
► To investigate factors that moderate or mediate the effects of musculoskeletal pain on health outcomes. For example, we will investigate whether specific social, physical and psychological factors play an intermediate role between low back pain and mobility decline.

In addition, we intend to use the OPAL Cohort to identify potential participants for future clinical trials of disability prevention in later life and to study disablement and multimorbidity more broadly. The 'cohort multiple randomised controlled trials' study design is becoming increasingly common.[18 19] The concept is to use data collected from an established cohort to identify people with specific health conditions and then, as and when the opportunity arises, invite them to participate in a clinical trial relevant to their condition.

In this paper, we describe the OPAL Cohort, design, data collection, and the profile of study participants at baseline and their overall representativeness of the English general population.

## COHORT DESCRIPTION
### Study design
A population-based, longitudinal, prospective cohort study in England, using a combination of annually administered, self-reported questionnaires and routinely collected health data.

### Practice and participant identification
#### General practice identification
General practices who were working with the National Institute for Health Research (NIHR) Clinical Research Network, which have been shown to be generalisable to wider primary care community,[20] were approached to take part in the study. In terms of geographical spread, we included a range of rural and urban areas across England, to capture diversity in both socioeconomic and ethnic profiles.

#### Participant identification
Eligible participants were identified from electronic record searches of general practice lists. A random sample of approximately 400 individuals (median: 365; range 158–400) per practice was selected (figure 1). To ensure an equal representation in two age bands: 65–74 and 75 years and over, around 200 individuals per practice within each age group were randomly selected.

#### Inclusion criteria
People registered with a general practice, aged 65 years and older, and living in the community, including sheltered or supported housing, were eligible for invitation.

#### Exclusion criteria
Individuals were excluded if they lived in a residential care or nursing home. Following the generation of the random sample, a designated general practitioner (GP) or research nurse from each practice screened the list to exclude those with known terminal illness with a life expectancy of less than 6 months, those who presented with severe health or social concerns sufficient to preclude approach, or those considered unable to provide informed consent.

### Sample size
The sample size was determined by the prevalence of lower back pain and musculoskeletal problems in older people and driven by the sample size requirement for the prognostic tool to assess mobility decline. We prespecified a minimum of 1000 participants of the sample should have lower back pain as this would be sufficient for a range of epidemiological analyses, including predictive modelling, within subsample of people with lower back pain.[21 22] The Cambridge Cohort Study of Ageing[23] provided the most recent estimates of disabling low back pain in the population aged 70–90 years, with prevalence

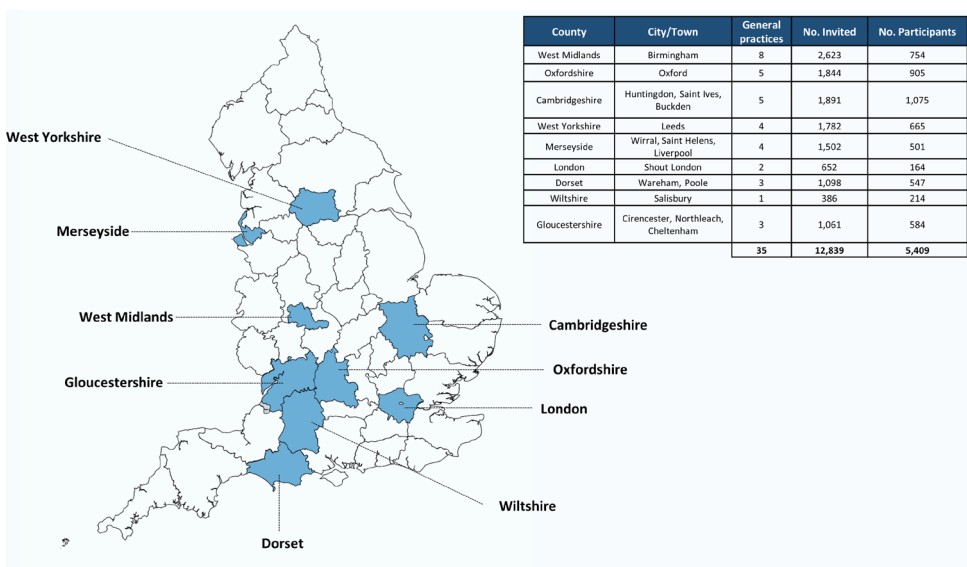

| County | City/Town | General practices | No. Invited | No. Participants |
|---|---|---|---|---|
| West Midlands | Birmingham | 8 | 2,623 | 754 |
| Oxfordshire | Oxford | 5 | 1,844 | 905 |
| Cambridgeshire | Huntingdon, Saint Ives, Buckden | 5 | 1,891 | 1,075 |
| West Yorkshire | Leeds | 4 | 1,782 | 665 |
| Merseyside | Wirral, Saint Helens, Liverpool | 4 | 1,502 | 501 |
| London | Shout London | 2 | 652 | 164 |
| Dorset | Wareham, Poole | 3 | 1,098 | 547 |
| Wiltshire | Salisbury | 1 | 386 | 214 |
| Gloucestershire | Cirencester, Northleach, Cheltenham | 3 | 1,061 | 584 |
| | | 35 | 12,839 | 5,409 |

**Figure 1** Locations of the areas from which the Oxford Pain, Activity and Lifestyle Cohort Study was derived. Map of England divided by counties.

of 25%–30% for these age groups, respectively. If we assume that 25% of people aged over 65 years have low back pain, then we required a minimum of 4000 people to yield 1000 with low back pain and 3000 people without low back pain. We estimated that between 30% and 40% of participants would agree to participate based on uptake to the Prevention of Falls Injury Trial,[24] which recruited an older population into an English falls prevention study and anticipated that there would be attrition from the sample over time. Therefore, we had to approach a minimum of 11 000 people, or approximately 350 people from each of 32 practices across our regions to achieve our recruitment target.

### Recruitment and enrolment
Recruitment and enrolment to OPAL commenced in October 2016 and completed in September 2018. A total of 12 839 individuals from 35 general practices in 9 different areas of England were invited to take part (figure 1). A pack including an invitation letter, participant information leaflet, consent form, baseline questionnaire and a postage paid return envelope was sent by the general practice. Five thousand four hundred and nine (42.1% of those eligible; range 5.1%–65.8% across practices) individuals who returned the baseline questionnaire and a signed consent form to the University of Oxford study office were enrolled in the study (figure 1). One-fifth (21.3% of those eligible; n=2736/12 839) declined participation and 4694 (36.6%; n=4694/12 839) did not respond. Non-responders were sent one postal reminder, 4 weeks after the original invitation. If no response was received, no further contact was made. The flow chart of the sample is illustrated in online supplementary figure S1.

### How often are participants followed up?
Study participants are followed up by postal questionnaire at annual intervals for 5 years. First year follow-up is completed, second and third year follow-up will complete in September 2020 and 2021, respectively. Future follow-up questionnaires will be sent at 4 and 5 years from the date of the original invitation.

### What is being measured?
#### Postal self-completed questionnaire
The OPAL Cohort Study includes information on a range of domains including demographic, socioeconomic, lifestyle variables, social participation, attitudes to ageing, musculoskeletal symptoms, health-related factors, comorbidity, mobility, disability, frailty, cognitive function, health-related quality of life (HRQoL) and medications (see table 1).

*Musculoskeletal symptoms* are assessed by asking the participant if they have experienced any trouble (ache, pain or discomfort) in nine different body sites (knees, hands/wrists, neck, shoulders, hips, feet/ankles, elbows, lower and upper back) during the last 6 weeks.[25 26] Information on presence, frequency, troublesomeness, onset, and description of back pain in the last 6 weeks was collected using recognised methods.[26–28] Information about the spread of back related symptoms was also included. To identify individuals with possible spinal stenosis we asked participants their pain travelled into their buttocks/legs, whether it was exacerbated while standing up or walking and whether the symptoms improved when sitting down or bending forward.[29 30] *Mobility* was assessed using different measures. Confidence to walk a half a mile was assessed using a single item from the Modified Gait Self-efficacy Scale, which is rated on a 1 'not confident at all' to 10 'totally confident' scale.[31] Participants also reported their perceived usual walking pace outdoors with six

**Table 1** Measures included in the OPAL Cohort Study

**Data collection for the OPAL Cohort Study**

| Domain measured | Self-reported measure | Years (Y) |
|---|---|---|
| Sociodemographic | Age, sex, education, relationship status | Y0–Y5 |
| | Participation in clubs and groups[54] | |
| | Requires unpaid/paid carer | |
| | Ethnicity | Y0 |
| | Number of live births and stillbirths | |
| Socioeconomic | Participant and GP area deprivation obtained from postcodes[42] | Y0–Y5 |
| | Current work status[55] | |
| | Type of housing | |
| | Adequacy of income[56] | |
| | Main occupation during lifetime[57] and self-rating of strenuousness of occupation | Y0 |
| | Internet access | |
| Lifestyle | Weight | Y0–Y5 |
| | Alcohol and smoking[58] | |
| | Current physical activity[59] | |
| | Height | Y0 |
| | Lifetime physical activity[60] | |
| General health data | Self-reported comorbidities and medication use | Y0–Y5 |
| | Sleep quality—Pittsburgh Sleep Quality Index[61] and average number of hours sleep each night | |
| | Incontinence—2 items from Barthel Index[62 63] | |
| | Falls in the last 12 months[33] | |
| | Broken bones or fractures in the last 12 months | |
| Musculoskeletal symptoms | The Nordic Musculoskeletal Questionnaire adapted version[25 26] | Y0–Y5 |
| | Report of back pain in last 6 weeks, troublesomeness, onset of back pain and nature of back pain[28] | Y0–Y5 |
| | Leg pain and symptoms related to low back pain | |
| | Screening questions for neurogenic claudication[29] | |
| | Report of knee pain, troublesomeness, interference with daily activity[64] | Y1–Y2 |
| | Location of knee pain | Y1 |
| Mobility | Change in mobility in the last year | Y0–Y5 |
| | Self-rated walking speed[65] | |
| | Use of walking aids (inside and outside) | |
| | Mobility concerns | |
| | Access to transport[54] | |
| | Life-Space Assessment[32] | |
| | Single item from the Modified Gait Self-efficacy Scale (10-item)[31] | |
| | Difficulty with balance while walking | Y2–Y5 |
| | Difficulties walking a half of mile[66] | Y3–Y5 |
| | Difficulties walking up and down a flight of stairs[66] | |

Continued

**Table 1** Continued

**Data collection for the OPAL Cohort Study**

| Domain measured | Self-reported measure | Years (Y) |
|---|---|---|
| Disability | Self-reported difficulty with activities of daily living (bathing, transfers, toilet use, dressing and eating) | Y2–Y5 |
| Frailty | Tilburg Frailty Index[34 35] | Y0–Y5 |
| Cognition | Clock-Drawing Test[67] | Y0–Y5 |
| Beliefs about ageing | Attitude to Ageing Questionnaire—physical changes subscale[68] | Y0–Y5 |
| Health-related quality of life | EuroQol 5-Dimension Health Questionnaire, five-level version[37] | Y0–Y5 |
| | EQ-VAS[37] | |

EQ-VAS, EuroQol Visual Analog Scale; GP, general practitioner; OPAL, Oxford Pain, Activity and Lifestyle.

possible responses: 'unable to walk', 'very slow', 'stroll at an easy pace', 'normal', 'fairly brisk' and 'fast'. Change in mobility in the last year was measured with the question 'Compared with 1 year ago, how would you rate your walking in general?' (Response options: much better, somewhat better, about the same, somewhat worse or much worse than a year ago). Participant, family, friends or doctor's concerns about participant ability to walk and move around were measured using two questions. Potential responses were 'extremely', 'a little concerned' or 'not concerned at all'. Life-space mobility was measured using five questions from the Life-Space Assessment (LSA) Questionnaire[32]: 'During the past 4 weeks have you gone to: (1) other rooms in your home besides the room where you sleep? (2) An area outside of your home as your porch, deck or patio, hallway or garage? (3) Different places in your neighbourhood? (4) Locations outside of your neighbourhood, but within your city? And (5) Places outside your town?' *Falls* data were collected as recommended by the Prevention of Falls Network Europe, using a single question, 'In the last 12 months, have you had any fall including a slip or trip following which you have come to rest on the ground, floor or lower level?'[33] Three possible responses were available: not fallen, fallen once or more than once in the last year. *Frailty* was measured by The Tilburg Frailty Indicator,[34 35] which is composed of two parts. The first part describes different determinants of frailty based on sociodemographic data and health-related questions. The second part contains 15 items, which measure three frailty domains: physical (8 items), psychological (4 items) and social (3 items). Frailty total scale and individual domain scores are derived from the second part. All items are rated as a binary response of either 0 or 1. Scores are the sum of the respective item points with a total score ranged from 0 to 15, with higher scores representing more frailty. A total score ≥5 points indicates that the individual is frail.[34] *Cognitive function* was measured with a Clock-Drawing Test.[36] Participants were asked to draw the entire face of a clock depicting the time '10 min after 11' following the instructions given in the questionnaire. Scoring was a six-point system

according to visual-spatial aspects and the correct denotation of time: normal cognition (score 6); minor visuospatial errors (score 5); mild (score 4), moderate (score 3) or severe (score 2) visuospatial disorganisation of time, or no reasonable representation of a clock (score 1).

*HRQoL* was measured by the EuroQol-5D-5L (EQ-5D-5L) Questionnaire, a generic measure of HRQoL that includes five levels of functioning from level 1 (no problems) to level 5 (severe or extreme problems).[37 38] Additionally, respondents rated their current health status according to the EuroQol Visual Analog Scale (EQ-VAS), from 0 (worst imaginable health) to 100 (best imaginable health). The responses from the five domains were converted into a single EQ-5D Index Value using the EQ-5D-5L Crosswalk Index Value Calculator to produce a final QoL Value.[39 40] The index values ranged between −0.594 (a state worse than death) and 1 (best possible health state).

New variables have been added to follow-up questionnaires (table 1), allowing the cohort to be used for a wider range of analytical approaches and purposes, and to dovetail to recruitment of new clinical trials. The first follow-up (Year 1) repeated baseline variables (table 1) with the exception of ethnicity, number of children, height, education, lifetime physical activity, main occupation during lifetime, self-rating of strenuousness of occupation and use of smartphone or computer to access the internet. Added variables included presence, frequency, troublesome, location and description of knee pain. The second wave of follow-up of data collection is collecting variables included in previous wave (Year 1) in addition to difficulty balancing while walking and any difficulty in the following basic activities of daily living; bathing, transfers, toilet use, dressing and eating. Each activity will be rated from 'no difficulty' to 'unable to perform'.

## Characteristics of participating general practices

General practice deprivation and estimated proportion of non-white ethnic groups in the practice population were obtained from Public Health England (PHE).[41] Deprivation was measured by the Index of Multiple Deprivation 2015 (IMD 2015).[42] Practice IMD scores are practice population weighted based on the Lower Layer Super Output Areas (LSOAs) where the practice population resides. LSOA is a low-level geography designed to contain 1500 inhabitants on average. Following the 2011 census, there were 32 844 English LSOAs.

General practice urbanity was defined using the 2011 urban–rural classification.[43] Within this classification, any settlement with a population of 10 000 people or more is defined as urban, with all others are classified as rural. It was determined at the LSOA level. Each general practice postcode was linked to its LSOA and it was then matched to urbanity.[44]

## Data management and quality control

All data are being processed and stored according to the Data Protection Act 2018. As the OPAL Study predated General Data Protection Regulation (GDPR) 2018, all participants were sent an updated GDPR statement along with their next annual questionnaire.

A software application was developed to support the filtering and random sampling of individuals from the practice lists. Identifiable data were removed by the application. When eligible participants were selected, a unique screening number was allocated to each participant and given to the practice. Each general practice put invitation letters into the corresponding prenumbered participant pack and completed the mail out.

The study office in Oxford receives returned questionnaires and the coordinating team undertake data quality checks. Returned questionnaires are processed using the electronic data capture software TeleForm Workgroup (Serial Number: 247885; Company name: ePartner Consulting), which includes internal system validation checks. Once questionnaires are scanned, additional validation is manually completed by a member of the OPAL Study team. For example, if a questionnaire is returned with a double-page spread missing, the participant is contacted by telephone with a maximum of two attempts (on 2 separate days) to complete missing sections.

## Access to electronic linkage

The majority of OPAL participants (99.2% of those who agreed to participate; n=5367/5409) consented for the research team access their UK National Health Service (NHS) Digital and primary healthcare records, and to be approached for future interventional and observational studies (up to date, data linkage are not completed). NHS Digital is a national provider of information, data and information technology systems for commissioners, analysts and clinicians in health and social care. The database holds information on hospital admissions, outpatient and accident and emergency department visits for individuals receiving NHS hospital treatment in England.[45] Diagnoses are coded using WHO's International Classification of Disease V.10. In addition, date and cause of death will be purchased/linked to NHS Digital.

## Patient and public involvement statement

Patients and the public were involved in the development of the research question, the design of the study and the conduct of the research. We piloted and refined the OPAL Cohort Study questionnaires with our Patient and Public Involvement (PPI) representatives. Our PPI group included older adults for whom English was a second language in order to ensure acceptability of wording of materials and to assist with uptake of the study by ethnic minority groups. We will continue to collaborate with our PPI representatives when drafting publications and with dissemination of findings to patients and the public.

## Statistical analysis

Descriptive statistics were used to summarise demographic and health-related measures of the OPAL participants at baseline. Selected key demographic and health-related variables are reported in this manuscript.

To assess whether our cohort is representative of the population of England, we compared a range of demographic and health-related characteristics of the OPAL Cohort Study with the 2011 England Census[46] and with The ELSA Cohort.[47] We deliberately focus on absolute differences and not on statistical significance because the large study samples may produce low p values even when absolute differences are small. Analyses were performed using STATA software V.15.1 (StataCorp).

### The English Longitudinal Study of Ageing

The ELSA Study is an ongoing prospective cohort study of a representative sample of community-dwelling people aged 50 years or older living in England.[47] It started in 2002 (wave 1), with participants recruited from an annual cross-sectional survey of households who were then followed up every 2 years. For this comparison, we used cross-sectional ELSA data from the core members (n=7223) at wave 8 (May 2016 to June 2017), as the time period was comparable with the OPAL Study on recruitment. ELSA participants aged <65 years (n=2102) and institutionalised (n=56) were excluded for the comparison. Thus, data from 5065 ELSA participants were included.

We compared the following participant characteristics between ELSA and OPAL: work status (retired vs non-retired), current relationship status (married vs non-married), weight, smoking status and health-related self-reported doctor-diagnosed chronic diseases (arthritis, diabetes, heart problems, stroke, dementia, lung disease, osteoporosis and high blood pressure).[48] We applied the recommended weightings to the data to correct for non-response in ELSA Cohort Study.[49]

Further details of the variables used in OPAL and ELSA Cohort Studies are described in online supplementary table S1. The ELSA data management is available in a Stata do-file 'online supplementary Data_management_ wave8_Dec2019' in supplementary information. The measurement protocol for the ELSA Cohort Study can be found online (http://www.ifs.org.uk/elsa).

### Dealing with missing data

Bias due to missing data (and the mechanism causing the data to be missing) will be investigated and an appropriate analysis approach, such as multiple imputation and/or inverse-probability weighting, to manage this problem will be used depending on the type of study being analysed. Only observed characteristics of OPAL participants at baseline are shown in this manuscript.

## FINDINGS TO DATE
### Response to invitation to participate

A total of 8145 individuals (63.4% among the 12 839 eligible participants) who were sent the invitation letter responded to the invitation, 5409 individuals (66.4% among the 8145 responders) agreed to participate in the study and 2736 individuals declined to participated (see online supplementary figure S1).

Age and sex distribution of participants and non-participants (declined and non-responders) are shown in online supplementary table S2, and by general practice in online supplementary figures S2 and S3. Overall, the participation rate in the OPAL Cohort Study was lower in the oldest age group (participation rates were over 44% for those aged 65–79 years and 36% for those aged 80+ years, respectively), although these were within the expected response rate. Response rate was similar between sexes (participation rates were 44.2% and 42.8% in men and women, respectively). No differences between participants and non-participants in terms of age or sex were observed, and these results were consistent across most practices.

Questionnaire response rate (among eligible individuals) by practice ranged from 5.1% to 65.8% (median: 45.6%; IQR: 32.2%–54.3%). Lower levels of response were observed in the most deprived practices (see online supplementary table S3).

OPAL baseline data have a low proportion of missing values. The amount of missing data for any single variable varied from 0.2% (n=13/5409) (for relationship status and current work status) to 5.9% (n=321/5409) (for Tilburg Frailty Score (0–15); item missing ranging from 0.4% to 1.9%).

### Characteristics of OPAL Study participants at baseline

The demographic characteristics of participants are reported in table 2. Half of the participants were women (51.5%; n=2784/5409), and the mean (SD) age was 74.9 (6.8) years, ranging from 65 to 100 years. The majority of study participants were white (94.9%; n=5132/5409).

The majority of participants were married or partnered (66.6%; n=3602/5409), with a higher proportion of women living alone. Most participants were retired (84.8%; n=4589/5409), and had secondary school education (56.4%; n=3051/5409). The median (IQR) area deprivation score of participants was 12.5 (6.9–20.3) and it was similar between sexes. In England, the median (IQR) deprivation score is 17.4 (9.7–30.1). Women were less likely to report that they were current smokers or drinking alcoholic beverages at least once every week than men. The prevalence of overweight (Body Mass Index (BMI): 25–29.9 kg/m$^2$) and obesity (BMI: ≥30 kg/m$^2$) among the whole sample was 38.1% (n=2061/5409) and 18.6% (n=1005/5409), respectively.

Health-related variables of men and women are described in table 3 and figure 2. A high proportion of OPAL participants (84.0%; n=4543/5409) reported musculoskeletal symptoms in at least one body area in the previous 6 weeks, with symptoms being more prevalent in women than men (table 3). Low back pain was the most frequently reported site for pain (44.4%; n=2404/5409).

The majority of participants were mobile and were confident to walk half a mile (66.1%; n=3577/5409), with a higher proportion of men being confident walkers.

**Table 2** Sociodemographic and life-style factors of men and women in the OPAL Cohort Study

| Characteristic | Men (n=2625) | Women (n=2784) |
|---|---|---|
| Age, mean (SD) | 74.8 (6.7) | 75.0 (6.8) |
| Age groups, n (%) | | |
| 65–69 | 784 (29.9) | 801 (28.8) |
| 70–74 | 696 (26.5) | 734 (26.4) |
| 75–79 | 542 (20.7) | 618 (22.2) |
| 80–84 | 355 (13.5) | 356 (12.8) |
| 85–89 | 196 (7.5) | 203 (7.3) |
| 90+ | 52 (2.0) | 72 (2.6) |
| Ethnicity (white), n (%) | 2465 (93.9) | 2667 (95.8) |
| Relationship status, n (%) | | |
| Married/civil union | 1897 (72.3) | 1506 (54.1) |
| Living with partner | 114 (4.3) | 85 (3.1) |
| Unmarried (never married) | 117 (4.5) | 105 (3.8) |
| Separated/divorced | 185 (7.1) | 273 (9.8) |
| Widow/widower | 305 (11.6) | 809 (29.1) |
| Live alone, n (%) | 534 (20.3) | 1021 (36.7) |
| Education, n (%) | | |
| High professional or university | 1017 (38.7) | 895 (32.2) |
| Secondary school only | 1370 (52.2) | 1681 (60.4) |
| None or primary | 219 (8.3) | 189 (6.8) |
| Work status (retired), n (%) | 2187 (83.3) | 2402 (86.3) |
| Quintiles of IMD, n (%) | | |
| Q1—Most deprived | 293 (11.2) | 289 (10.4) |
| Q2 | 323 (12.3) | 339 (12.2) |
| Q3 | 542 (20.7) | 613 (22.0) |
| Q4 | 575 (21.9) | 591 (21.2) |
| Q5—Least deprived | 892 (34.0) | 952 (34.2) |
| BMI (kg/m$^2$), mean (SD) | 26.8 (4.3) | 26.4 (5.3) |
| Smoking status, n (%) | | |
| Never | 1071 (40.8) | 1618 (58.1) |
| Ex-smoker | 1401 (53.4) | 1040 (37.4) |
| Current | 145 (5.5) | 118 (4.2) |
| Cigarettes per day, median (IQR) | 15 (10–20) | 10 (5–17) |
| Alcohol intake once per week, n (%) | 1861 (70.9) | 1361 (48.9) |

Data included older adults 65 years and older at baseline 2016–2018.
BMI, Body Mass Index; IMD, Index of Multiple Deprivation; OPAL, Oxford Pain, Activity and Lifestyle.

Over one-third (38.7%; n=2094/5409) of participants rated their walking speed as strolling at an easy pace or very slow, 18.5% (n=1002/5409) reported using a walking aid inside or outside and 25.5% (n=1375/5409) reported

**Table 3** Health-related characteristics of men and women at the OPAL Cohort Study

| Health-related characteristics | Men (n=2625) | Women (n=2784) |
|---|---|---|
| Musculoskeletal symptoms in the last 6 weeks, n (%) | | |
| Low back (small of the back) | 1098 (41.8) | 1306 (46.9) |
| One of both knees | 932 (35.5) | 1132 (40.7) |
| Wrist/hands | 653 (24.9) | 1053 (37.8) |
| Neck | 673 (25.6) | 951 (34.2) |
| Shoulders | 667 (25.4) | 948 (34.1) |
| One of both hips/thighs | 599 (22.8) | 875 (31.4) |
| One or both ankles/feet | 559 (21.3) | 755 (27.1) |
| Upper back | 160 (6.1) | 346 (12.4) |
| Elbows | 161 (6.1) | 173 (6.2) |
| Any pain, n (%) | 2137 (81.4) | 2406 (86.4) |
| Mobility | | |
| Confidence to walk half a mile, median (IQR) | 10 (9–10) | 10 (6–10) |
| Outdoor walking pace, n (%) | | |
| Fast | 91 (3.5) | 93 (3.3) |
| Fairly brisk | 534 (20.3) | 572 (20.6) |
| Normal | 994 (37.9) | 958 (34.4) |
| Stroll at an easy pace | 647 (24.7) | 726 (26.1) |
| Very slow | 326 (12.4) | 395 (14.2) |
| Unable to walk | 19 (0.7) | 27 (1.0) |
| Walking rate than 1 year ago, n (%) | | |
| Much better | 52 (2.0) | 84 (3.0) |
| Somewhat better | 114 (4.3) | 101 (3.6) |
| About the same | 1822 (69.4) | 1831 (65.8) |
| Somewhat worse | 507 (19.3) | 622 (22.3) |
| Much worse | 113 (4.3) | 133 (4.8) |
| Walking aid use inside (yes), n (%) | 108 (4.1) | 153 (5.5) |
| Walking aid use outside (yes), n (%) | 306 (11.7) | 435 (15.6) |
| Falls in the last year, n (%) | | |
| None | 1900 (72.4) | 1906 (68.5) |
| One fall | 474 (18.1) | 624 (22.4) |
| More than one fall | 235 (9.0) | 236 (8.5) |
| Frailty, Tilburg Frailty Score, median (IQR) | 2 (1–4) | 3 (1–5) |
| Clock-Drawing Test, n (%) | | |
| 1 point | 9 (0.3) | 5 (0.2) |
| 2 points | 28 (1.1) | 45 (1.6) |
| 3 points | 112 (4.3) | 102 (3.7) |
| 4 points | 210 (8.0) | 273 (9.8) |

| Table 3 Continued | | |
|---|---|---|
| Health-related characteristics | Men (n=2625) | Women (n=2784) |
| 5 points | 445 (17.0) | 487 (17.5) |
| 6 points | 1756 (66.9) | 1793 (64.4) |
| Quality of life | | |
| EQ-5D Crosswalk Index Value, mean (SD) | 0.79 (0.19) | 0.76 (0.21) |
| EQ-VAS, mean (SD) | 79.1 (16.7) | 77.7 (18.0) |

Sample sizes may vary due to missing values; data included older adults 65 years and older at baseline 2016–2018.
EQ-VAS, EuroQol Visual Analog Scale; IQR, Interquartile Range; OPAL, Oxford Pain, Activity and Lifestyle.

that their walking speed to be slower than a year ago. Over a quarter of participants (29.0%; n=1569/5409) reported having fallen once or more in the 12 months prior to the baseline questionnaire, and 27.1% (n=1463/5409) were classified as frail. Frailty was more prevalent in women. The majority of study participants presented high cognitive function, with 82.8% (n=4481/5409) of participants having a score of 5 or 6 points in the Clock-Drawing Test. Most of the participants reported good health across four domains of the EQ-5D-5L questions with 88.5% (n=4784/5409), 69.7% (n=3772/5409), 66.1% (n=3577/5409) and 59.0% (n=3190/5409) reporting no problems with self-care, anxiety/depression, usual activities and mobility, respectively, except for pain/discomfort with a percentage of participants reporting no problems of 29.5% (n=1594/5409). The average HRQoL measured by EQ-5D-5L Crosswalk Value set and the EQ-VAS were 0.79 (SD 0.20) and 78.4 (SD 17.4), respectively. Women reported worse HRQoL (lower average score in both scales) compared with men (table 3). The average self-reported EQ-VAS Score in population norms for UK population aged 65–74 and 75 years and over[50] is broadly comparable with the OPAL Study (population norm vs OPAL Study: 77.3 vs 80.5 and 73.8 vs 75.6, respectively).

The more frequently self-reported health condition was high blood pressure (45.5%; n=2459/5409), followed by arthritis (44.2%; n=2391/5409) and angina or heart problems (20.2%; n=1094/5409). High blood pressure

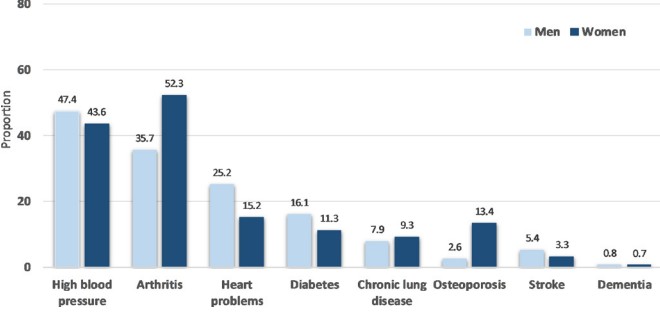

**Figure 2** Health conditions in men and women of Oxford Pain, Activity and Lifestyle Cohort Study.

was the most prevalent condition among men (47.4%; n=1244/2625), and arthritis the most prevalent in women (52.3%; 1455/2784) (figure 2).

### Representativeness of OPAL Cohort Study
Demographic characteristics in OPAL Cohort Study were similar to the general population of the same age range in the 2011 England Census (see online supplementary table S4). There was a lower proportion of women in the 80 and older age group in OPAL Study compared with the general population.

Online supplementary tables S5 and S6 show the sex-specific distribution of health-related characteristics in OPAL and ELSA Cohort Studies across four age groups. Overall, health-related characteristics of the OPAL participants were broadly comparable with those recruited to the nationally representative ELSA Cohort Study.

Both men and women participants in the OPAL Study were less likely to smoke and had a lower prevalence of self-reported heart problems, stroke and dementia.

### Characteristics of included general practices
General practice area deprivation and the estimated proportion of ethnic groups registered in the practice population are described in online supplementary table S3. Of the 35 general practices included in the study, 32 had data available on PHE national general practice profiles website. Overall, 9 of 32 practices (28.1%) were classified among the most deprived practices (IMD deciles 1–3), 14 of 32 (43.8%) in the most affluent practices (IMD deciles 8–10) and the remainder categorised as moderate (n=9/32; 28.1%; IMD deciles 4–7).

Over 14.3% (n=5/35) of general practices are located in rural areas, a slightly lower proportion than across rural areas in England as a whole (17.0%; n=5598/32 844 LSOAs).

### Cohort multiple randomised controlled trial
The first registered randomised controlled trial using the OPAL Cohort Study is now being undertaken. This NIHR-funded trial is testing the effectiveness of a physiotherapist delivered combined physical and psychological intervention for older adults with neurogenic claudication compared with best practice advice (BOOST).[51] The trial is registered with the International Standard Randomised Controlled Trials Database, reference number ISRCTN12698674.

### STRENGTHS AND LIMITATIONS
The original target for recruitment of the OPAL Cohort Study was a minimum of 4000 older adults from 32 general practices. However, uptake was better than predicted and we have recruited 5409 older adults from 35 general practices within 9 distinct areas, providing good geographical coverage within England. The wide range of self-report health measures will allow us to account for a large range of potential mediating and confounding variables.

One important limitation of the cohort is the reliance on self-reported data. We acknowledge that performance tests may provide more reliable objective data; however, we were interested in patient reported factors and outcomes as these are feasible to capture during a patient consultation and findings may be applied within clinical practice. We also have obtained written informed consent to access NHS Digital and primary healthcare data for the majority of the participants, to allow independent verification of diagnoses related to hospital admission and attendance, and as well as important elements of health service resource use and mortality. Biological markers are not systematically collected in electronic health records and this may be a potential weaknesses. However, the OPAL Cohort Study was designed to elucidate the epidemiology of musculoskeletal pain and the contribution of pain on health-related outcomes rather than attempt to investigate the biological underpinning of musculoskeletal pain.

Individuals living in more deprived neighbourhoods (based on practice population deprivation) and non-white ethnicity groups were less likely to participate in OPAL (see online supplementary table S3). This finding is consistent with other epidemiological studies which report that populations with a lower socioeconomic position are less likely to take part in research compared with those with higher socioeconomic position.[52] Nevertheless, our population is broadly representative of the English population.

Our findings will apply to community-dwelling older adults in England and may under represent those living in the community with severe cognitive impairment.

In terms of the representativeness of the OPAL Study, demographic and health-related characteristics of OPAL participants are similar to those in the general population (2011 Census) and ELSA Study, respectively, (see online supplementary tables S4–S6). The selected variables for the comparison analysis had good comparability in both OPAL and ELSA Studies, but there were some differences. For example, in ELSA, weight was calculated using measured weight, whereas in OPAL weight was self-reported. Self-reported weight tends to be underreported, particularly by women and those who are heaviest.[53] In addition, in ELSA, the definition of 'smoker status' and health conditions combines information from previous waves, whereas in OPAL Study, only baseline information was used. This may have led to a slight underestimation of the difference between ELSA and OPAL in the percentage of 'ex-smoker' and individuals with the health condition.

## FUTURE WORK

Data collection for the Year 1 Follow-up Questionnaire was completed in September 2019 and Year 2 and 3 Follow-up Questionnaires will be completed in 2020 and 2021, respectively. We plan to administer questionnaires at annual intervals, and aim to continue this for a minimum of 5 years.

The potential of this data set has yet to be exploited and further work is in progress. We will start focusing on particular health domains (such as low back pain and mobility problems), together with an exploration of factors underlying the variability of those health domains. For example, we will investigate whether social, physical and psychological factors mediate the effect between low back pain and immobility. Future work will include the development of a prognostic tool to identify older adults at risk of mobility decline to help individuals, GPs and other health professionals identify risk factors and when these should be prioritised as a treatment target. This longitudinal cohort study will also identify health trajectories and will examine their associations with demographic, clinical and social factors, with the aim of identifying factors that maintain good health and independence in older people.

## COLLABORATION

We welcome potential collaborations with other research groups. Interested researchers should contact Professor Sarah (Sallie) Lamb (S.E.Lamb@exeter.ac.uk/sarah.lamb@ndorms.ox.ac.uk) to discuss collaboration. Further information on the OPAL Cohort Study can be found on our website (http://www.ndorms.ox.ac.uk/rrio/opal).

**Author affiliations**
[1]Centre for Rehabilitation Research in Oxford, Nuffield Department of Orthopaedics, Rheumatology and Musculoskeletal Sciences, University of Oxford, Oxford, UK
[2]Warwick Clinical Trials Unit, Division of Health Sciencies, University of Warwick, Coventry, UK
[3]Department of Sport, Exercise & Rehabilitation, Northumbria University, Newcastle-Upon-Tyne, UK
[4]Shool of Primary, Community and Social Care, Keele University, Keele, UK
[5]College of Medicine and Health, University of Exeter, Exeter, UK

**Acknowledgements** The authors would like to thank the individuals who participated in The Oxford Pain, Activity and Lifestyle Cohort Study, the general practitioners and their staff for assisting with the identification and invitation of eligible individuals and the National Institute for Health Research Biomedical Research Centre, Oxford for funding this study (grant reference PTC-RP-PG-0213-20002).

**Collaborators** OPAL Study Team: Conway O, Darton F, Dutton S, Hagan D, Haywood D, Hewitt A, Marian I, Nevay L, Nicolson P, Slark M, Vadher K, Watson M, Williamson E, Arden N, Barker K, Collins G, Fairbank J, Fitch J, French D, Griffiths F, Hanson Z, Hutchinson C, Petrou S. OPAL General Practice Team: Grange Hill Surgery, Birmingham; Gosford Hill Medical Centre, Oxford; River Brook Medical Centre, Birmingham; The Key Medical Practice, Oxford; Summertown Health Centre, Oxford; Alconbury and Brampton Surgeries, Cambridgeshire; Old Exchange Surgery, Cambridgeshire; The Wand Medical Centre, Birmingham; Kingsfield Medical Centre, Birmingham; Buckden and Little Paxton Surgery, Cambridgeshire; Temple Cowley Medical Group, Oxford; Keynell Covert Surgery, Birmingham; Burbury Medical Centre, Birmingham; Hollow Way Medical Centre, Oxford; Priory View Medical Centre, Leeds; Newton Surgery, Leeds; Priory Fields Surgery, Cambridgeshire, Cromwell Place Surgery, Cambridgeshire; Craven Road Medical Centre, Leeds; Queslett Medical Practice, Birmingham; Hall Street Medical Centre, Saint Helens; Vauxhall Health Centre, Liverpool; Ireland Wood Surgery, Leeds; Civic Medical Centre, Wirral; Brownlow Group Practice, Liverpool; Wareham Surgery, Dorset; The Adam Practice, Poole; The Harvey Practice, Dorset; Three Chequers Medical Practice, Salisbury; Gate Medical Centre, Birmingham; Rendcomb Surgery, Cirencester; Cotswold Medical Practice, Cheltenham; Brigstock and South Norwood

Partnership, Croydon; Portland Practice, Gloucestershire; Eversley Medical Centre, Croydon. Supporting the National Institute for Health Research Clinical Research Networks: Thames Valley and South Midlands, Eastern; Yorkshire and the Humber, North West Coast; Wessex, West of England; West Midlands, South London.

**Contributors** MTSS participated in the data preparation, analysis and interpretation; and the development and writing of the paper. EW participated in the Oxford Pain, Activity and Lifestyle (OPAL) Study design, data collection and interpretation of the results of the paper. JB, LW and CDM participated in the OPAL Study design, data collection and interpretation of findings. AG and AM participated in the design of the OPAL Study, data collection, data preparation and interpretation of findings. SEL conceived the study, secured funding and oversaw all aspects as principal investigator and participated in the design and execution of the OPAL Study, and the development and writing of the paper. All authors contributed and approved the final manuscript.

**Funding** This research is funded by the National Institute for Health Research (NIHR) Programme Grants for Applied Research (reference: PTC-RP-PG-0213-20002). Preparatory work for the programme of research was supported by the NIHR Collaborations for Leadership in Applied Health Research and Care. CDM is funded by the NIHR Applied Research Collaboration (West Midlands), the NIHR School for Primary Care Research and an NIHR Research Professorship in General Practice (NIHR-RP-2014-04-026). JB is supported from NIHR Research Capability Funding via University Hospitals Coventry and Warwickshire.

**Map disclaimer** The depiction of boundaries on the map(s) in this article do not imply the expression of any opinion whatsoever on the part of BMJ (or any member of its group) concerning the legal status of any country, territory, jurisdiction or area or of its authorities. The map(s) are provided without any warranty of any kind, either express or implied.

**Competing interests** All authors have completed the Unified Competing Interest form at www.icmje.org/coi_disclosure.pdf. JB reports grant funding from the National Institute for Health Research (NIHR), Diabetes Research UK and Medtronic. CDM reports grants from NIHR, during the conduct of the study. SEL reports and declared competing interests of authors: SEL was on the Health Technology Assessment (HTA) Additional Capacity Funding Board, HTA End of Life Care and Add-on Studies Board, HTA Prioritisation Group Board and the HTA Trauma Board. All other authors declare no conflicts of interest.

**Patient and public involvement** Patients and/or the public were involved in the design, or conduct, or reporting, or dissemination plans of this research. Refer to the Methods section for further details.

**Patient consent for publication** Not required.

**Ethics approval** Ethical approval for the study was provided by the London—Brent Research Ethics Committee (16/LO/0348) on 10 March 2016. All participants provided written informed consent, returned with the baseline questionnaire before being enrolled in the cohort study.

**Provenance and peer review** Not commissioned; externally peer reviewed.

**Data availability statement** Further information on the Oxford Pain, Activity and Lifestyle Cohort Study can be found on our website: https://www.ndorms.ox.ac.uk/rrio/opal. Unpublished data will be available for data sharing. Enquires can be made to Professor Sarah (Sallie) Lamb (Principal Investigator, e-mail: sarah.lamb@ndorms.ox.ac.uk / S.E.Lamb@exeter.ac.uk).

**ORCID iDs**
Maria T Sanchez Santos http://orcid.org/0000-0003-1908-8623
Esther Williamson http://orcid.org/0000-0003-0638-0406
Julie Bruce http://orcid.org/0000-0002-8462-7999

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
