## [Reviewer comments · BMJ Open]

ARTICLE DETAILS

TITLE (PROVISIONAL)	Cohort profile: Oxford Pain, Activity and Lifestyle (OPAL) study, a prospective cohort study of older adults in England
AUTHORS	Sanchez Santos, Maria; Williamson, Esther; Bruce, Julie; Ward, Lesley; Mallen, Christian; Garrett, Angela; Morris, Alana; Lamb, Sarah

VERSION 1 – REVIEW

REVIEWER	Christophe J Bula University of Lausanne Medical Center, Switzerland
REVIEW RETURNED	01-Mar-2020

GENERAL COMMENTS	Overall appreciation: This is a very well written paper that provides a detailed description of the design of this new cohort study. Results of baseline characteristics of participants and the potential of future analyses are clearly presented, although this part could have been expanded. In particular, the authors should better delineate how and to what extent this new cohort will add to already existing cohorts of older persons in England (i.e., ELSA), as well as in other developed countries (e.g. In-Chianti, Baltimore, etc.). General comments: This cohort profile paper describes the OPAL cohort study that aims at investigating the contribution of musculoskeletal pain as well as other factors on subsequent mobility decline and disability in later life among community-dwelling older persons aged 65 and over. This very well written paper offers a thorough and detailed description of the design, enrollment criteria and recruitment process, follow-up methods, as well as data selection and management of the OPAL cohort study. All methodological and analytical steps are described in a detailed and clear manner, and they appear completely appropriate. Results show a good response rate for this type of cohort (42.1%), and few missing data at baseline. Results from the comparison of characteristics in OPAL and ELSA participants suggest that, despite some differences (higher proportion of men in the 80+ age group in OPAL), most socio-demographics are broadly comparable. The comparisons of health conditions' prevalence across the two cohorts however suggest that OPAL participants are on average in better health than the ELSA representative sample. The authors appropriately acknowledge this limitation and attribute it to self-report bias. The relatively large sample size and the diversity of practices, including those in deprived area are clear strengths. Another clear strength of this cohort is the large proportion of participants who agreed to consent to access their UK NHS Digital and primary health care data. There are however some limitations in this cohort's design. The first, acknowledged by the authors, is the reliance only on self-reported data. In-person performance test as well as systematic collection of biomarkers (e.g., blood and neuroimaging) would definitely have enhanced the potential of this particularly well-designed cohort study. Access to UK NHS Digital and primary health care data will not address this issue unless some performance test are
--

	systematically collected in UK across England (e.g., cognitive screening test ?, frailty assessment ?). Otherwise, only tests decided by the GP, resp the hospital team, will be available. An additional limitation relates to the – not unexpected - lower response rate in participants from deprived practice. Finally, the exclusion of participants unable to consent is another concern, as older persons with cognitive impairment will likely be underrepresented. This is further suggested by the lower prevalence of dementia in OPAL compared to ELSA participants (Suppl Table S4). This should be acknowledged by the authors (see also comment below). Specific comments: Title No specific comment. Abstract: No specific comment. Strengths and limitations: 1) P3: The authors should acknowledge here that the exclusion of individuals unable to provide consent likely resulted in the exclusion of older persons with cognitive impairment thus likely precluding inference from their results to this specific population (see also comment below). Introduction: 2) P4, L22-23: The rationale behind the creation of the OPAL cohort could be further strengthened. The authors provide a list of well-known associations between musculoskeletal pain and adverse health events (e.g., falls) as well as trajectories (incidence of frailty, cognitive decline, etc.). On the other hand, they refer to gaps in current knowledge on the health consequences of pain. Could the authors provide to the reader one or two examples of specific gaps they identified in knowledge about the link between musculoskeletal pain and health trajectories and outcomes. It seems to me that examples related to the investigation of some specific moderators and mediators of pain's effect on health trajectories will likely be most original. Cohort description: 3) P6, L20-23: Individuals considered unable to provide informed consent were excluded. Could the authors comment on this decision as this likely resulted in underrepresentation of individuals with cognitive impairment, a clear limitation to the generalizability of their future findings to this important population. Findings to date: No specific comment Strengths and limitations: 4) P16, L18-55: See previous comments above (i.e., selection bias among cognitively impaired persons). Future work: 5) P17, L5-30: As previously mentioned, the potential of this new cohort study to address specific issue about the consequences of pain on health trajectories could be better delineated. Tables and supplementary tables: No specific comment. Figures: No specific comment. References 19) P21, L15: The correct reference of ref#4 should be Ageing Soc 2014;34:452-71
--	--

REVIEWER	Yao He Institute of Geriatrics, Beijing Key Laboratory of Aging and Geriatrics, National clinical research center for geriatrics diseases, State Key Laboratory of Kidney Disease, Second Medical Center of Chinese PLA General Hospital, 28 Fuxing Road, Beijing, 100853, China
-----------------	---

REVIEW RETURNED	25-Apr-2020
-------------

GENERAL COMMENTS	The cohort aims to study the determinants of later life health by investigating the situation of Pain, Activity and Lifestyle in the elderly. Unfortunately, they identified some major concerns, especially regarding the design of your study and the interpretation of the data.  1. The strength part does not explain the outstanding strengths of this study, so it is suggested to refine it again (Page 3, Lines 6-28) 2. Please supplement the method of sample size calculation. 3. Please explain in more detail the meaning of "General Practice who were predominantly working with the UK NIHR Clinical Research Network" (CRN), and its impact of sampling according to General Practice on representativeness. For example, what proportion of rural areas and urban areas it covers, how many are the number of General Practice who were predominantly working with CRN, what's the proportion in the total, and whether the proportion in different sizes of urban and rural areas is similar. (Page 6, Lines 35-42) 4. Suggest to add a flow chart to show the process of participant recruitment. 5. Please supplement the specific process of sampling and randomization implementation. Besides, the part of "Participant identification" mentioned that each practice is sampled according to 65-74 years and 75 years and over stratification to ensure representativeness. Please add the specific proportion of stratification and its basis. (Page 6, Lines 47-60) 6. Please provide criteria for overweight and obesity. 7. It is suggested to describe the end point and duration of follow-up. 8. Please supplement a discussion about the characteristics of the lost population among the eligible population and the bias that it may cause. 9. In Supplemental Table S2, S3, and S4, the data formats of OPAL cohort and ELSA cohort are inconsistent, please modify it or explain the reason. 10. It is suggested to provide both absolute difference and statistical p value in the statistical analysis (Page 13, Lines 6-11) 11. It is recommended that all domains of the questionnaires measured at the baseline and follow-up stages should be presented in tabular form (Pages 8-10) 12. Please clarify the significance and purpose of Elsa cohort included in this manuscript.
--

REVIEWER	Christine McGarrigle Trinity College Dublin, the University of Dublin, Ireland.
REVIEW RETURNED	10-May-2020

GENERAL COMMENTS	This manuscript describes the study design of the Oxford Pain, activity and lifestyle study. The cohort is a prospective longitudinal study of community living adults aged over 65, investigating the determinants of health in later life. The cohort intends to be a source of recruitment for clinical trials, in addition to answering research questions. This is a well-written paper that describes the cohort clearly, I have some comments below for clarification some suggested additions. Cohort description  1. The study design is to use a combination of self-reported questionnaire and routinely collected health data. While Table 1 describes what is collected in the questionnaire, there is no detailed description of what health data, and data through other linkages, will be available. The high acceptance for data linkage of the cohort is a real strength of this cohort, and the paper would benefit from some further detail and discussion on potential data that will be available through data linkage. This should be added either to Table 1, or as an additional table. 2. Two response rates are presented, one with total eligible as the denominator (abstract, Figure 1 and Table S5), and one with total responded as the denominator (Response, page 14 line 16, (65.8%). This should be made into a figure so its clearer for the reader, a flow diagram explaining numbers approached, included and reasons for non-participation if known. A standard term for either response rate/participation rate/refusal rate, should then be explained and used consistently. 3. In addition, the age, ethnicity, sex and educational attainment, if possible, should be compared for those who responded and those who didn't, or at least for those who responded compared to those who refused. While a participation rate alone doesn't necessarily determine the extent of bias, readers should be provided with as much other information as possible to allow them to make a judgement on that. According to Supplementary table S5, the nonresponse is
---

	socio-economically graded, and higher in less affluent GP practices. It would be informative to present this both as non-response due to refusals, and overall non-participation, i.e. non-contact as they may have different implications. 4. The cohort has a 42% participation rate, which is low and may raise concerns of non-participation bias. This must be addressed in more detail in the paper, particularly as the reasons for study participation may be associated with the outcomes of interest. Some discussion is required about the implications of low recruitment from high deprivation index GPs on the cohort representativeness. Given that disability is known to be socially patterned both in the UK and worldwide, this might have serious implications for inferences that can be drawn from the study and the research questions of interest to the paper. Details of how the study has considered and will deal with these issues of potential nonresponse bias should be discussed in the paper, and what steps will be taken to mitigate this. For example, will inverse probability weights be developed for non-response, based on the discrepancies between the proportions in the sample and the proportions in the population to take this into account and weights can be calculated to increase the importance of respondents who are under-represented in the data. While the cohort may be not designed to be a nationally representative cohort, it should still address differential inclusion in the study in order to allow the findings to be applicable to other populations. Population weights are generated for most longitudinal studies. These can then include attrition weights to account for differential attrition as the cohort waves progress. 5. The results of the manuscript report a prevalence of 82% reporting pain, 83% in women, however in the introduction, Page 5 line 10, results from a recent review reports prevalence of chronic pain ranged from was 42% to 72%, consistent with other countries. This difference between the current cohort and why this might be higher than other studies should be discussed. 6. Page 13 line 4 states "The mean deprivation score of individuals (SD) was 16.6 (14.1) and it was similar between sexes". How does this deprivation score compare to the general population? It may answer the questions in point 4 above if it is similar to the general population. 7. Representativeness of OPAL cohort study. On page 3, line 28, it states that the cohort participants are similar to those in the general population. Page 16, line 6 discusses this and tables S2, S3 and S4, however, it is unclear why the cohort was not compared to the population of England for demographics rather than just ELSA as the ELSA study must be weighted to the Census population to account for differential response and attrition. While it may be necessary to compare to ELSA for the health outcomes, a comparison should also be provided for the census population of England. Supplemental table 2 this should include the age and sex distribution of the general population from census data and a comparison for the ethnicity and educational attainment distributions would also be beneficial. 8. The strengths and limitations of the study bullet points, page 3, only include strengths, this should be expanded to include limitations.
--	---

VERSION 1 – AUTHOR RESPONSE

Reviewer 1

Reviewer Name: Dr. Christophe J Büla

Institution and Country: University of Lausanne Medical Center, Switzerland

This is a very well written paper that provides a detailed description of the design of this new cohort study. Results of baseline characteristics of participants and the potential of future analyses are clearly presented, although this part could have been expanded.

1. In particular, the authors should better delineate how and to what extent this new cohort will add to already existing cohorts of older persons in England (i.e., ELSA), as well as in other developed countries (e.g. In-Chianti, Baltimore, etc.).

Author response: We thank the reviewer for this comment.

OPAL cohort is a new cohort specifically targeting older people in England. Although there are many other cohort studies examining age-related health conditions among community dwelling older adults e.g. the English Longitudinal Study of Ageing, the MOBILIZE Boston Study, the Longitudinal Study of Ageing, the Baltimore Longitudinal Study of Aging and the Invecchiare in Chianti, aging in the Chianti area (InChianti), to our knowledge, only one cohort has focused on the impact and contribution of musculoskeletal pain on mobility decline and disability: the MOBILIZE Boston Study.¹ This American cohort study is ongoing but has a relatively small sample size (n=765) compared to OPAL cohort.

Author action: We have now added the following paragraph in the introduction section to page 5, lines 86-92:

“There are a number of high quality cohort studies examining age-related health conditions among community dwelling older adults. These include the English Longitudinal Study of Ageing, the MOBILIZE Boston Study, the Longitudinal Study of Ageing, the Baltimore Longitudinal Study of Aging and the Italian Invecchiare aging in Chianti study (InChianti), amongst many others. However, to our knowledge, only one cohort focuses on the impact and contribution of musculoskeletal pain on disability in older people, the ongoing MOBILIZE Boston Study. This American cohort is limited by a relatively small sample size (765 participants at inception).”

General comments:

This very well written paper offers a thorough and detailed description of the design, enrollment criteria and recruitment process, follow-up methods, as well as data selection and management of the OPAL cohort study. All methodological and analytical steps are described in a detailed and clear manner, and they appear completely appropriate.

We thank the reviewer for these supportive comments

2. There are however some limitations in this cohort's design. The first, acknowledged by the authors, is the reliance only on self-reported data. In-person performance test as well as systematic collection of biomarkers (e.g., blood and neuroimaging) would definitely have enhanced the potential of this particularly well-designed cohort study. Access to UK NHS Digital and primary health care data will not address this issue unless some

performance test are systematically collected in UK across England (e.g., cognitive screening test ?, frailty assessment ?). Otherwise, only tests decided by the GP, resp the hospital team, will be available.

Author response: We agree with the reviewer that this study is somewhat limited by use of self-report data and we have acknowledged this in the limitations section of the manuscript. This is a large population cohort from across England and although we recognise the potential for inclusion of biomarkers, our aim was to design a cohort study to elucidate the epidemiology of MSK and the contribution of pain on health related outcomes rather than attempt to investigate the biological underpinning of MSK pain.

Additionally, regarding our main health outcome (mobility), it is acknowledged that different aspects of mobility are captured when comparing self-reported mobility and physical performance measures. Numerous studies have compared subjective and objective measurement of mobility and most report a moderate correlation between the two approaches²⁻⁴. Also, objective physical performance measures do not capture important mobility-related psychological factors, such as confidence to mobilise independently or fear of movement (kinesophobia). Self-reported mobility outcomes are crucially important as they reflect the individual's perceptions about their own mobility and independence. This is important when considering mobility trajectory over time as psychological factors may contribute as much as physical factors. For example, fear of falling has been shown to predict falls. We acknowledge that biomarkers provide additional health information but it was not feasible to incorporate within this large-scale epidemiological design.

Our study is novel in that it includes multiple self-reported variables based on several important models of ageing and disablement⁵. We are interested in frailty, falls, mobility and disability as these factors are easy to assess during a face-to-face patient consultation and our findings will inform clinical practice.

Author action: We have added the following sentences in the discussion as limitations of this study to page 20, lines 433-443:

“One important limitation of the cohort is the reliance upon self-reported data. We acknowledge that performance tests may provide more reliable objective data, however, we were interested in patient reported factors and outcomes as these are feasible to capture during a patient

consultation and findings may application within clinical practice. We also have obtained written informed consent to access NHS Digital data...”

“...Biological markers are not systematically collected in electronic health records and this may be a potential weakness. However, the OPAL cohort study was designed to elucidate the epidemiology of musculoskeletal pain and the contribution of pain on health related outcomes rather than attempt to investigate the biological underpinning of musculoskeletal pain.”

3. An additional limitation relates to the – not unexpected - lower response rate in participants from deprived practice.

Author response: Thanks for this comment; we agree with the reviewer. As expected, the response

rate from participants invited from more deprived practices was lower than those living in more wealthy areas. This finding is consistent with other epidemiological studies which report that populations with a lower socioeconomic position are more likely to decline participation in research studies⁶.

Author action: We have included the following sentence in the ‘Strengths and limitations’ section (page 20, lines 444-449):

“Individuals living in more deprived neighbourhoods (based on practice population deprivation) and non-white ethnicity groups were less likely to participate in OPAL (Supplementary Table S3). This finding is consistent with other epidemiological studies which report that populations with lower socioeconomic position are less likely to take part in research compared to those with higher socioeconomic position⁶. Nevertheless, our population is broadly representative of the English population.

4. Finally, the exclusion of participants unable to consent is another concern, as older persons with cognitive impairment will likely be underrepresented. This is further suggested by the lower prevalence of dementia in OPAL compared to ELSA participants (Suppl Table S4). This should be acknowledged by the authors (see also comment below).

Author response: The prevalence of dementia is very low in the community which makes estimation of precise levels difficult in any cohort. Within OPAL we rely on dementia having been diagnosed by the doctor, and the participant telling us. Our

estimates for participants telling us a doctor has diagnosed dementia is lower than expected in comparison to the weighted ELSA sample, notably in the oldest old. We have now inserted additional data on the Clock Drawing Test which is a measure of executive function into the paper, and report a higher prevalence of executive impairment than ELSA⁷. However, methods of testing are different between the two cohorts and the Clock Drawing Test is very sensitive to minor levels of impairment in executive function⁸. We have acknowledged in the revised manuscript that our estimates of dementia are uncertain and may be an underestimate.

Author action: We have acknowledged this in ‘Strengths and limitations’ section (page 20, lines 450-451):

“Our findings will apply to community-dwelling older adults in England and may under represent those living in the community with severe cognitive impairment”.

Specific comments:

Strengths and limitations:

5. P3: The authors should acknowledge here that the exclusion of individuals unable to provide consent likely resulted in the exclusion of older persons with cognitive impairment thus likely precluding inference from their results to this specific population (see also comment below).

Author response: For details, please refer to our response to comment number 4.

Introduction:

6. P4, L22-23: The rationale behind the creation of the OPAL cohort could be further strengthened. The authors provide a list of well-known associations between musculoskeletal pain and adverse health events (e.g., falls) as well as trajectories (incidence of frailty, cognitive decline, etc.). On the other hand, they refer to gaps in current knowledge on the health consequences of pain. Could the authors provide to the reader one or two examples of specific gaps they identified in knowledge about the link between musculoskeletal pain and health trajectories and outcomes. It seems to me that examples related to the investigation of some specific moderators and mediators of pain’s effect on health trajectories will likely be most original.

Author response: We thank the reviewer for this comment. We agree with the reviewer that the OPAL cohort study will address some specific gaps in knowledge.

Mediation and moderation have recently received increased attention. A previous systematic review and meta-analysis on possible psychological, social and physical factors to explain the effect of pain on disability in people with LBP and neck pain⁹ reported a total of 37 mediator analyses (12 studies). The age of included participants ranged from 31 to 51 years and over half (60%) were male. The majority of studies (n=7; 22 models) were cross-sectional and the data from the remaining longitudinal studies (n=8; 15 models) were of poor quality. Therefore, a better understanding of the causal path between pain and disability in representative community-based older adults is needed to inform decisions about treatment and rehabilitation. The OPAL cohort study will help to better understand the causal path between musculoskeletal pain and mobility decline in older adults, with detailed hypotheses on the pathways linking social, physical and physiological factors with onset of mobility change.

Author action: We have clarified how this study will address gaps in knowledge. In the introduction section, we have also added examples of moderators of the effect of pain on health trajectories (page 4, lines 81-85; page 5, lines 101-102).

Cohort description:

7. P6, L20-23: Individuals considered unable to provide informed consent were excluded. Could the authors comment on this decision as this likely resulted in underrepresentation of individuals with cognitive impairment, a clear limitation to the generalizability of their future findings to this important population.

Author response: For details, please refer to our previous response to comment number 4. Our findings will apply to community-dwelling older adults in England and may under represent those with severe cognitive impairment.

Strengths and limitations:

8. P16, L18-55: See previous comments above (i.e., selection bias among cognitively impaired persons).

Author response: Thanks for your suggestions. Please refer to our response to your previous comment number 4

Future work:

9. P17, L5-30: As previously mentioned, the potential of this new cohort study to address specific issue about the consequences of pain on health trajectories could be better delineated.

Author response: Thanks for your suggestions. We have now added the following paragraph in future work (page 21, lines 469-471):

“For example, we will investigate whether social, physical and psychological factors mediate the effect between low back pain and immobility”

References

10. P21, L15: The correct reference of ref#4 should be Ageing Soc 2014;34:452-71 PubMed

Author response: We thank the reviewer for bringing this to our attention

Author action: The reference has now been corrected

Reviewer 2

Reviewer Name: Dr. Yao He

Institution and Country: Institute of Geriatrics, China.

1. The strength part does not explain the outstanding strengths of this study, so it is suggested to refine it again (Page 3, Lines 6-28)

Author response: Thank you very much for this comment. We agree with the reviewer that the strengths of this study were not well described in the introduction.

Author action: We have now refined the introduction and also expanded section on strengths and limitations. Please refer to sections introduction (pages 4-5, lines 81-92) and Strengths and Limitations (pages 19-20).

2. Please supplement the method of sample size calculation.

Author response: Thanks for your suggestion.

Author action: We have added the sample size calculation to the main manuscript in the methods section (page 7, lines 143-156)

3. Please explain in more detail the meaning of “General Practice who were predominantly working with the UK NIHR Clinical Research Network” (CRN), and its impact of sampling according to General Practice on representativeness. For example, what proportion of rural areas and urban areas it covers, how many are the number of General Practice who were predominantly working with CRN, what’s the proportion in the total, and whether the proportion in different sizes of urban and rural areas is similar. (Page 6, Lines 35-42)

Author response: The Clinical Research Network’s role is to facilitate health and care organisations to participate in high quality research by supporting research activities such as screening and patient recruitment. General practices working with the CRN are “research active” and, therefore, likely to want to participate when approached about a study. In England there were approximately 7,000 general practices and 48% of them were actively engaged in clinical research during the period from April 2016 to March 2017 and a previous study in the UK has shown that research active practices are not substantially different to those not participating in research¹⁰.

OPAL participating practices comprise 8 out of 15 (53.3%; 8/15) Local Clinical Research Networks (LCRNs) that, together, cover the whole of England (North West Coast, Yorkshire and Humber, West Midlands, West of England, Thames Valley and Shout Midlands, Eastern, Wessex and South London).

The proportion of OPAL participating practices located in rural areas is 14.3% (5/35 LSOAs where general practices are located) and this is similar to the proportion of rural areas in England as a whole (17.0%; 5,598/32,844 English LSOAs). Urbanity is defined using the 2011 urban-rural classification and collected through the 2011 census¹¹. Settlements with a population of 10,000 people or more are defined as urban, all others are classified as rural. Data were obtained from the Office for National Statistics open geography portal¹².

Additionally, a description of the sociodemographic indicators for participating practices were already given in ‘Characteristics of included general practices’ and Supplementary information Table S3, where area deprivation and ethnicity groups based on the practice population resides are shown.

Author action: We have deleted the word ‘predominantly’ and added the following sentence in the ‘General practice identification’ section (Page 6, line 122):

“General practices who were working with the NIHR Clinical Research Network (CRN), *which have been shown to be generalisable to wider primary care community*⁰, ...”

We have also provided general practice urbanity information in 'What is being measured-Characteristics of participating general practices' (Page 11, lines 245-250) and 'Findings to date-Characteristics of included general practices' (Page 19, lines 417-418) sections:

"General practice urbanity was defined using the 2011 urban-rural classification¹¹. Within this classification, any settlement with a population of 10,000 people or more is defined as urban, with all others are classified as rural. It was determined at the LSOA level. Each general practice postcode was linked to its LSOA and it was then matched to urbanity¹²."

"Over 14.3% (n=5/35) of general practices are located in rural areas, a slightly lower proportion than across rural areas in England as a whole (17.0%; n=5,598/32,844)."

4. Suggest to add a flow chart to show the process of participant recruitment.

Author response: Thanks for this comment.

Author action: A flow chart with information of the sample in the OPAL study has now been added to the supplementary information (See Supplementary Figure S1).

5. Please supplement the specific process of sampling and randomization implementation. Besides, the part of "Participant identification" mentioned that each practice is sampled according to 65-74 years and 75 years and over stratification to ensure representativeness. Please add the specific proportion of stratification and its basis. (Page 6, Lines 47-60)

Author response: We selected participants from the practice lists using a random sampling frame that selected an equal number of people aged 65-74 years and 75 years and older. We selected age 75 and over to represent the oldest old. As anticipated not as many people aged 75 years and older accepted the invitation to participate, but ultimately the sample was broadly representative of the age and sex mix of the older population of England. As in the unweighted ELSA sample, OPAL has a very slight underrepresentation of the older women. We will develop, explore and, if it makes a meaningful difference, publish a weighting algorithm to address this in future analyses.

Author action: We have now clarified this section in the main manuscript. Please see 'Participant identification' section (Page 6, lines 126-130)

6. Please provide criteria for overweight and obesity.

Author response: The criteria for overweight was a BMI between 25 and 29.9 kg/m² and for obesity BMI≥30 kg/m²

Author action: this criteria was added in brackets in the results section (Page 17, line 366).

7. It is suggested to describe the end point and duration of follow-up.

Author response: We plan to administer questionnaires at annual intervals, and aim to continue this for a minimum of five years. This information was included in 'Future work' section.

Author action: This information has been included in the 'How often are participants followed up?' section as well (Page 8, line 172).

8. Please supplement a discussion about the characteristics of the lost population among the eligible population and the bias that it may cause.

Author response: Only information about age and sex amongst the eligible population but, who declined participation, was available. Age and sex distribution between individuals who did and did not participate were compared. Participants who were 80 years and older were less likely to participate in OPAL (Supplementary Table S2, Figure S1-S2), however, on average, participants had similar age to those of non-participants.

Author action: A Supplementary Table S2 and two Figures S1 and S2 have been now added to supplementary information with information on demographic characteristics for participants and non-participants (divided into declined and non-responders) for overall sample and by general practice.

This paragraph has been included in 'Findings to date-Response to invitation to participate' section (Page 16, lines 340-347):

"Age and sex distribution of participants and non-participants (declined and non-responders) are shown in Supplementary Table S2, and by general practice in Supplementary Figure S2-S3. Overall, the participation rate in the OPAL cohort study was lower in the oldest age group (participation rates were over 40% for those aged 65-79 years and 36% for those aged 80+ years, respectively), although these were within the expected response rate. Response rate was similar between sexes (participation rates were 44.2% and 42.8% in men and women, respectively). No

differences between participants and non-participants in terms of age or sex was observed, and these results were consistent across most practices.”

In addition, the following paragraph has been added in ‘Strength and limitations’ section (Page 20, lines 444-449):

“Individuals living in most deprived neighbourhoods (based on practice population deprivation) and non-white ethnicity groups were less likely to participate in OPAL (Supplementary Table S2-S3). This is consistent with other epidemiological studies, which report that populations with a lower socioeconomic position are less likely to take part in research compared to those with higher socioeconomic position⁶. Nevertheless, our population is broadly representative of the English population.”

9. In Supplemental Table S2, S3, and S4, the data formats of OPAL cohort and ELSA cohort are inconsistent, please modify it or explain the reason.

Author response: Supplemental Table S2, S3, and S4 showed the sex-specific distribution of characteristics in OPAL and ELSA cohort studies across four age groups. The distribution of characteristics in OPAL cohort study correspond to the observed data and in ELSA to the estimated data, so data in the ELSA cohort study were weighted to correct for non-response. In future analyses, we will develop, explore and, if it makes a meaningful difference, publish a weighting algorithm to address this.

Author action: None

10. It is suggested to provide both absolute difference and statistical p value in the statistical analysis (Page 13, Lines 6-11)

Author response: We deliberately focus on absolute differences and not on statistical significance because the large study samples may produce low p-values even when absolute differences are small. It was mentioned in the ‘Statistical analysis’ section.

Author action: None

11. It is recommended that all domains of the questionnaires measured at the baseline and follow-up stages should be presented in tabular form (Pages 8-10)

Author response: Measures included in the OPAL cohort study at baseline and follow up are shown in Table 1.

Author action: None

12. Please clarify the significance and purpose of Elsa cohort included in this manuscript.

Author response: The ELSA cohort study is based on a representative sample of non-institutionalized people aged 50 and older in England (similar than OPAL cohort study, but our age range was those aged 65 and over). The ELSA sample was refreshed at Wave 7 from the Health Survey for England (HSE) 2011-2012 and the latest wave collected was at 2016-2017 (Wave 8), so the time frame was equivalent to timing of the baseline data collection for the OPAL study.

We decided to use ELSA cohort study for comparison for the following reasons: 1) ELSA cohort study was designed to be a representative sample of older people living in the community in England and one of its waves was at 2016-2017 (Wave 8), equivalent to the baseline data of the OPAL cohort study, 2) detailed health-related information was not available for the 2011 Census population of England. However, we have now compared demographic characteristics (age, sex and ethnicity) of the OPAL study to those in the general population using the most recent and available census data in 2011 for England. We observed that age, sex and ethnicity distributions for the OPAL cohort study were almost identical to the general population. A small underrepresentation of women with age 80 and older was observed.

Author action: We have now compared demographic characteristics of participants in the OPAL cohort study with the general population in England 2011, we have added it as a Supplementary Table S4.

The following paragraph is now included in 'Representativeness of OPAL Cohort study' section (Page 18, line 398-400):

“Demographic characteristics in OPAL cohort study were similar to the general population of the same age range in the 2011 England Census (Supplementary Table S5). There was a lower proportion of women in the 80 and older age group in OPAL study compared to the general population.”

Reviewer: 3

Reviewer Name: Dr. Christine McGarrigle

Institution and Country: Trinity College Dublin, the University of Dublin, Ireland

This is a well-written paper that describes the cohort clearly, I have some comments below for clarification some suggested additions.

Thanks for the supporting comment.

Cohort description

1. The study design is to use a combination of self-reported questionnaire and routinely collected health data. While Table 1 describes what is collected in the questionnaire, there is no detailed description of what health data, and data through other linkages, will be available. The high acceptance for data linkage of the cohort is a real strength of this cohort, and the paper would benefit from some further detail and discussion on potential data that will be available through data linkage. This should be added either to Table 1, or as an additional table.

Author response: Thank you very much for your comment. We agreed with the reviewer that data linkage of the cohort is one of the strength of the cohort, but unfortunately we have not completed data linkage as yet.

Author action: We have now added this sentence in the 'Access to electronic linkage' section (Page 13, lines 276-277): *"(Up to date, data linkage are not completed)"*

2. Two response rates are presented, one with total eligible as the denominator (abstract, Figure 1 and Table S5), and one with total responded as the denominator (Response, page 14 line 16, (65.8%). This should be made into a figure so its clearer for the reader, a flow diagram explaining numbers approached, included and reasons for non-participation if known. A standard term for either response rate/participation rate/refusal rate, should then be explained and used consistently.

Author response: Thank you. We presented the response rate for those who were eligible to participate and also for those who responded to the invitation. However, we agree with the reviewer that it may be confusing to the reader, so a flow diagram with that information is now added to supplementary information. Unfortunately, we were not permitted to collect data on reasons for non-participation by the ethics committee, therefore, these data are not included into the chart.

Author action: A flow chart with information of the sample in the OPAL study has now been added to the supplementary information (See Supplementary Figure S1).

3. In addition, the age, ethnicity, sex and educational attainment, if possible, should be compared for those who responded and those who didn't, or at least for those who responded compared to those who refused. While a participation rate alone doesn't necessarily determine the extent of bias, readers should be provided with as much other information as possible to allow them to make a judgement on that. According to Supplementary table S5, the nonresponse is socio-economically graded, and higher in less affluent GP practices. It would be informative to present this both as non-response due to refusals, and overall non-participation, i.e. non-contact as they may have different implications.

Author response/action: Thank you. Data on age and sex for eligible participants and deprivation and ethnicity based on general practices population (Supplementary Table S3 in the new version of the manuscript) were available. For further details, please refer to our response on Dr. Yao He's comment on eligible population question-8.

4. The cohort has a 42% participation rate, which is low and may raise concerns of non-participation bias. This must be addressed in more detail in the paper, particularly as the reasons for study participation may be associated with the outcomes of interest. Some discussion is required about the implications of low recruitment from high deprivation index GPs on the cohort representativeness. Given that disability is known to be socially patterned both in the UK and worldwide, this might have serious implications for inferences that can be drawn from the study and the research questions of interest to the paper. Details of how the study has considered and will deal with these issues of potential nonresponse bias should be discussed in the paper, and what steps will be taken to mitigate this. For example, will inverse probability weights be developed for non-response, based on the discrepancies between the proportions in the sample and the proportions in the population to take this into account and weights can be calculated to increase the importance of respondents who are under-represented in the data. While the cohort may be not designed to be a nationally representative cohort, it should still address differential inclusion in the study in order to allow the findings to be applicable to other populations. Population weights are generated for most longitudinal studies. These can then include attrition weights to account for differential attrition as the cohort waves progress.

Author response: We thank to the reviewer for raising this interesting comment. The response rate was comparable to a recent cohort studies in older adults^{13 14}. However, comparison of response rates across studies can be problematic without considering differences in sample design, eligibility criteria and fieldwork protocol.

Overall, the participation rate in the OPAL cohort study was lower in the oldest age group (participation rates were over 40% in those aged 65-79 years and 36% in those aged 80+ years, respectively), but it was within the expected response rate we estimated a priori. Response rate was very similar between sexes (participation rates were 44.2% and 42.8% in men and women, respectively). There was a lower participation rate in more deprived practices compared to those living in most wealthy areas, but it was not unexpected. This is consistent with other epidemiological studies, which report that populations with a lower socioeconomic status are more likely to decline to participate in research⁶.

Selection bias may arise from three mechanisms: (i) unwillingness to participate, (ii) missing information in some covariates (and thus, exclusion from some analyses) and (iii) attrition of the cohort (dropouts or losses to follow-up)

This bias would be a major concern only if the relationships between exposures and outcomes differed systematically in those who participated in OPAL and those who did not, and several studies have demonstrated that this is minimal and hence not a major concern^{6 15-17}. However, we cannot be sure that those results can be generalized to all exposures and outcomes.

Therefore, we will explore possible mechanisms underlying this bias and analysis approached, such as multiple imputation and/or inverse-probability weighting, to manage this problem will be examined depending of the type of study being analysis.

Author action: The age and sex distribution of participants and non-participants (declined and non-responders) are now shown in Supplementary Table S2, and divided by practice in Supplementary Figure S2-S3. We have also described the new findings in the 'Findings to date' (page 16, lines 340-347).

We have also added a new section 'Dealing with missing data' under 'Statistical analysis' section describing how we will deal with nonresponse and attrition bias (page 15, lines 329-332):

"Bias due to missing data (and the mechanism causing the data to be missing) will be investigated and an appropriate analysis approach, such as multiple imputation and/or inverse-probability weighting, to manage this problem will be used depending

of the type of study being analysed. Only observed characteristics of OPAL participants at baseline are shown in this manuscript”

5. The results of the manuscript report a prevalence of 82% reporting pain, 83% in women, however in the introduction, Page 5 line 10, results from a recent review reports prevalence of chronic pain ranged from was 42% to 72%, consistent with other countries. This difference between the current cohort and why this might be higher than other studies should be discussed.

Author response: We thank the reviewer for bringing this to our attention. Methodological differences probably explain the difference with the prevalence of chronic pain reported in a recent review. The prevalence reported in our study was measured by asking to the participant if they had trouble (ache, pain, discomfort). It was at any time in the past 6 weeks, therefore, it was different in comparison to other cohorts, that they tend to say 'on most days' or 'lasting longer than X weeks' or 'rated as 4/10 or greater'. We have now replaced “pain” with “musculoskeletal symptoms” throughout main manuscript.

However, the percentage of participants reporting any problem with pain measured by EQ5D pain was around 70%, which is consistent with the prevalence reported in the recent review and this is mentioned in ‘Findings to date’ section (page 17, line 385).

Author action: The label name ‘*Pain/ache/discomfort in the last 6 weeks*’ was added in the main manuscript Table 3 and “pain” has been replaced with “musculoskeletal symptoms” throughout the main manuscript.

6. Page 13 line 4 states “The mean deprivation score of individuals (SD) was 16.6 (14.1) and it was similar between sexes”. How does this deprivation score compare to the general population? It may answer the questions in point 4 above if it is similar to the general population.

Author response: Participants in the OPAL cohort study live in more wealthy areas compared to the general population (the median (IQR) of area deprivation score is 17.4 (9.7-30.1) in England and 12.5 (6.9-20.3) in the OPAL cohort study). However, this finding was not unexpected. The figure below shows the distribution of the individual deprivation score in OPAL against England.

Author action: We have now described the distribution of the individual deprivation score in OPAL and in England in the 'Findings to date' section (page 17, line 364). Further details about how we are going to deal with bias due to non-response are given in comment number 4.

7. Representativeness of OPAL cohort study. On page 3, line 28, it states that the cohort participants are similar to those in the general population. Page 16, line 6 discusses this and tables S2, S3 and S4, however, it is unclear why the cohort was not compared to the population of England for demographics rather than just ELSA as the ELSA study must be weighted to the Census population to account for differential response and attrition. While it may be necessary to compare to ELSA for the health outcomes, a comparison should also be provided for the census population of England. Supplemental table 2 this should include the age and sex distribution of the general population from census data and a comparison for the ethnicity and educational attainment distributions would also be beneficial.

Author response: Thank you very much for this relevant comment. We decided to use ELSA cohort study for the following reasons: 1) ELSA cohort study was designed to be a representative sample of the non-institutionalized people aged 50 and older in England and one of its waves was at 2016-2017 (Wave 8), equivalent to the baseline data of the OPAL cohort study, 2) detailed health-related information was not available for the 2011 Census population of England. However, we have now compared demographic characteristics (age, sex and ethnicity) of the OPAL study to those in the general population using the most recent and available census data in 2011 for England. We observed that age and ethnicity distributions for the OPAL cohort study were similar to the general population. A small underrepresentation of women with age 80 and older was observed.

Author action: We have edited Supplementary Table S2, and it now contains age and ethnicity distributions by sex for OPAL and 2011 Census population of England, and we have amended this new information in the statistical analysis data (page 14, lines 302-303) and findings to date section with the new data (page 18, lines 398-400).

8. The strengths and limitations of the study bullet points, page 3, only include strengths, this should be expanded to include limitations.

Author response: We thank the reviewer for bringing this to our attention

Author action: We have now added the following limitations of the study (page 3, lines 58-61):

- *“The cohort study relies on self-reported and routine NHS data, there is not face to face data collection”*
- *“Our findings may under represent older adults living in the community with severe cognitive impairment”*

VERSION 2 – REVIEW

REVIEWER	Christophe J Bula University of Lausanne Medical Center, Switzerland
REVIEW RETURNED	01-Jul-2020

GENERAL COMMENTS	The authors did an excellent job, their detailed answers addressed appropriately the queries.
---

REVIEWER	Yao He Chinese PLA general hospital, China
REVIEW RETURNED	07-Jul-2020

GENERAL COMMENTS	From my point of view, the work is well-done and provides cohort profile about Oxford Pain, activity and lifestyle, and the author fully responds to the comments and suggestions of the reviewers. Thus it merits to be published.
---

REVIEWER	Christine McGarrigle Trinity College Dublin, the University of Dublin
REVIEW RETURNED	13-Jul-2020

GENERAL COMMENTS	Thank you for the opportunity to re-review the manuscript. The authors have responded to all my points for clarification. I have no further comments on this revision.
--